# Induction of Hepcidin Expression in the Renal Cortex of Sickle Cell Disease Mice

**DOI:** 10.3390/ijms241310806

**Published:** 2023-06-28

**Authors:** Asrar Ahmad, Namita Kumari, Nowah Afangbedji, Sergei Nekhai, Marina Jerebtsova

**Affiliations:** 1Center for Sickle Cell Disease, Howard University, Washington, DC 20059, USA; asrar.ahmad@howard.edu (A.A.); namita.kumari@howard.edu (N.K.); nowah12@gmail.com (N.A.); snekhai@howard.edu (S.N.); 2Department of Microbiology, Howard University, Washington, DC 20059, USA; 3Departments of Medicine, Howard University, Washington, DC 20059, USA

**Keywords:** sickle cell disease, renal hepcidin, renal ferroportin, renal iron accumulation, inflammation

## Abstract

In patients with sickle cell disease (SCD), chronic hemolysis and frequent blood transfusions cause iron overload and accumulation in the kidneys. The iron deposition is found in the renal cortex and correlates with the severity of hemolysis. In this study, we observed a significant accumulation of iron in the renal cortex of a mouse model of SCD, and assessed the expression of the proteins involved in maintaining renal iron homeostasis. Despite the intracellular iron accumulation, the levels of the transferrin receptor in the kidneys were increased, but the levels of the iron exporter ferroportin were not altered in SCD mice. Ferroportin is regulated by hepcidin, which binds to it and promotes its degradation. We found reduced serum hepcidin levels but increased renal hepcidin production in SCD mice. Furthermore, we observed significant macrophage infiltration and increased expression of intercellular adhesion molecule 1 in the endothelial cells of the kidneys in SCD mice. These observations correlated with elevated levels of proinflammatory cytokines IL-1β and IL-6, which can potentially stimulate hepcidin expression. Taken together, our results demonstrate that in individuals with SCD, a renal inflammation state induces renal hepcidin production that blocks the upregulation of ferroportin levels, resulting in dysregulation of iron homeostasis in the kidney and iron deposition in the renal cortex.

## 1. Introduction

Iron (Fe) is an essential element for cell metabolism, yet excessive Fe can be detrimental due to the formation of highly reactive radicals catalyzed by Fe, which can cause tissue damage [1]. Because of their low solubility and high toxicity, Fe levels are tightly regulated by a group of specialized molecules responsible for Fe acquisition, transport, and storage in a non-toxic form to meet the cellular and organismal needs of the body [2]. The concentration of serum Fe is precisely balanced between its absorption in the small intestine, storage in the liver, recycling from senescent red blood cells (RBCs) in the spleen by macrophages, and retention in the kidneys. Circulating Fe is bound to transferrin (Tf), making up a small pool of approximately 3 mg that is highly dynamic and turns over more than 10 times daily [3]. About 70% of Fe in the human body is bound to hemoglobin within RBCs and erythroid progenitor cells [4]. A substantial amount of intracellular Fe in the body, up to 1 g, is stored in ferritin in the liver, while muscles typically have around 300 mg of Fe, mostly in the form of myoglobin, with all other body tissues (apart from the duodenum) containing about 8 mg of Fe [4]. The cellular Fe uptake involves the binding of Fe-loaded Tf to the Tf receptor (TfR1), followed by its endocytosis [5]. Inside endosomes, Fe (III) is released from Tf after acidification and is then reduced to Fe (II). This is followed by Fe (II) crossing the endosomal membrane via the divalent metal transporter 1 (DMT1). Intracellular Fe is mostly used in mitochondria for the synthesis of heme and Fe-S cluster proteins, and it incorporates cytosolic metalloproteins. The excess of intracellular Fe is stored in ferritin or exported by ferroportin (FPN) [6]. Ferritin is ubiquitously expressed in the cytosol of all cells and is composed of 24 heavy (H) and light (L) subunits that together form a shell-like nanocage for Fe storage. This structure can accommodate up to 4500 Fe atoms in the form of ferric oxyhydroxide phosphate [6].

In chronic systemic Fe overload, Tf becomes saturated, leading to the formation of a toxic form of Fe (non-Tf-bound Fe—NTBI) in plasma [7]. Compared to Tf-bound Fe, NTBI has a more than 300 times higher rate of cell uptake, causing the accumulation of non-ferritin-bound Fe (labile Fe) in the cells [8].

Chronic hemolysis and blood transfusions in patients with hemoglobin disorders could lead to systemic Fe overload and the accumulation of Fe in organs. The pattern of Fe accumulation within organs differs in β-thalassemia and sickle cell disease (SCD). Cardiac Fe accumulation is usually found in β-thalassemia patients, not SCD patients [9,10,11]. Cardiac complications such as heart failure and arrhythmias caused by “Fe-induced” cardiomyopathy are the leading cause of death in β-thalassemia patients [9]. In contrast, renal cortical Fe deposition is more common in SCD than in other hematologic disorders [10]. Interestingly, in comparison to homozygous SCD patients, patients with sickle cell/β-thalassemia compound heterozygous mutation have a significantly lower frequency of renal Fe overload, despite similar levels of Fe present in the liver, heart, and pancreas [12]. In SCD patients, liver Fe accumulation correlates with blood transfusion amount, while kidney Fe accumulation is related to the severity of hemolysis [13,14]. The relationship between Fe in the circulation and Fe accumulation in organs is not fully understood.

Fe metabolism is regulated systemically and at the cellular level. The Fe response elements (IRE) and Fe response protein (IRP) system control cellular Fe metabolism [15], while systemic Fe homeostasis is regulated by hepcidin (HAMP), a liver-produced peptide hormone. Fe, inflammation, and endoplasmic reticulum stress stimulate HAMP transcription, while anemia, hypoxia and erythropoiesis inhibit it [16]. HAMP binds to the cellular Fe exporter, FPN, causing its degradation, resulting in reduced Fe uptake in the duodenum and increased intracellular Fe accumulation. Patients with SCD and β-thalassemia have low levels of circulating HAMP due to anemia [17,18]. Thus, cardiac Fe accumulation in β-thalassemia patients and renal Fe accumulation in SCD patients are unlikely to be driven by the circulating HAMP. Recent studies have indicated that extrahepatic HAMP expression can control intracellular Fe levels in peripheral organs [19,20,21,22]. HAMP expression has been observed in the heart [19], kidney [20], brain [21], and placenta [22]. Additionally, endogenously expressed HAMP plays an essential role in the cardiomyocyte Fe homeostasis and cardiac Fe accumulation in β-thalassemia [23].

The kidney is involved in Fe recycling, preventing Fe loss in the urine. Circulating Tf-bound Fe is filtered in the renal glomerulus, before binding to TfR1 and being reabsorbed in the proximal renal tubules through endocytosis. The Tf-bound Fe is then released from the TfR1 in the acidified endosomes and exported into the cytoplasm by DMT1. Intracellular Fe is either stored bound to the ferritin or released from cells by FPN, located at the basolateral plasma membrane of the proximal tubular cells [24,25]. Two ferroxidases, hephaestin (Heph) and ceruloplasmin (CP), facilitate Fe release from FPN [26]. HAMP is expressed in the renal distal tubule cells in healthy humans, mice, and rats [20]. In cases of Fe overload, the kidney is exposed to elevated levels of Tf-bound Fe and NTBI, but the metabolism of Fe in the kidney has received little consideration compared to Fe metabolism in the liver, pancreas, and heart.

We hypothesized that increased expression of HAMP in the renal cortex plays a key role in Fe accumulation in SCD. To test this hypothesis, we used the Townes SCD mouse model (SCD mice) [27]. We demonstrated here that elevated HAMP production in the renal cortex of SCD mice reduced FPN levels in the kidney but not in the liver. Higher levels of renal HAMP were associated with increased renal inflammation and the renal expression of pro-inflammatory cytokines IL-1β and IL-6, which induced HAMP production in the kidney.

## 2. Results

### 2.1. Iron Accumulation in the Renal Cortex of SCD Mice

Blood was collected from six SCD mice (Townes) and six control mice through the retroorbital vein plexus. Hematological parameters were assessed using a veterinary hematology counter Sysmex XN-1000 with C57BL/6 chip as described in Materials and Methods Section 4.2 (Table 1). Serum and urine Fe concentrations were assessed by an Fe Assay kit (Abcam, Cambridge, UK) as described in the Materials and Methods Section 4.2. Total Fe binding capacity (TIBC) was measured by mouse TIBC ELISA kit (BioLegend, San Diego, CA, USA), and Tf saturation (TSAT) was calculated (Materials and Methods, Section 4.2). Blood was collected from both male and female mice (three of each sex). There were no statistically significant differences between hematological parameters of male and female mice (Appendix A).

Blood and urine samples were collected from six SCD and six control mice (three males and three females for each group). Hematological parameters were assessed using the Sysmex XN-1000 hematology counter with C57BL/6 chip. Serum and urine Fe concentrations were measured by an Fe Assay kit (Abcam, Cambridge, UK). Total Fe binding capacity (TIBC) was measured using the mouse TIBC ELISA kit (BioLegend), and TSAT was calculated as described in the Materials and Methods Section 4.2. Urine was collected using metabolic cages, and urine creatinine concentration was assessed using QuantiChrom™ Creatinine Assay Kit (BioAssay Systems). GFR was assessed using a transdermal fluorescent detector (MediBeacon GmbH, St. Louis, MO, USA) of sinistrin-FITC.

The combined results of six mice (three males and three females) are shown in Table 1. There are no standard protocols for hematological and urinary parameters in mice, and the results are strongly dependent on the protocol and vary between different laboratories. In Table 1, the reference range was combined using results from two reviews [28,29] and the mouse phenome database at the Jackson Laboratory (https://phenome.jax.org/strains/7, accessed on 7 June 2023). Most of the parameters in the control mice were in the reference range (Table 1). The levels of red blood cell counts (RBC) and mean corpuscular volume (MCV) were lower, but the levels of mean corpuscular hemoglobin (MCH) were higher than the reference range. SCD mice exhibited signs of hemolysis and anemia, including low hematocrit, hemoglobin, and RBC count, as well as increased levels of circulating reticulocytes (Table 1). The levels of these parameters were lower than in control mice but still within the reference range. MCV was significantly increased in SCD mice compared to control mice, reflecting RBC sickling (Table 1, 40.02 ± 1.46 in SCD vs. 33.28 ± 0.6 in control, *p* = 1.34 × 10^−5^), while MCH was not significantly different (Table 1, 9.33 ± 0.12 in SCD vs. 9.07 ± 0.29 in control, *p* = 0.2943). This suggested that reduced levels of hemoglobin were likely associated with fewer RBCs, not a decrease in the amount of hemoglobin per cell. Serum Fe levels were higher in SCD mice (Table 1, 165.11 ± 38.79 µg/dL in SCD vs. 113.81 ± 33.37 µg/dL in control, *p* = 0.0618). Unexpectedly, TIBC demonstrated a trend toward a higher level in SCD mice (Table 1, 532.83 ± 99.71 µM in SCD vs. 321.55 ± 149.12 µM in control, *p* = 0.05663). This was contrary to expectation, as TIBC is usually higher in Fe deficiency, when serum Tf concentrations are elevated and serum Fe concentrations are reduced [30]. We did not find a significant difference in the TSAT between SCD and control mice (Table 1, 34.51 ± 9.08% in SCD vs. 37.81 ± 12.29% in control, *p* = 0.6814).

Urine was collected using metabolic cages, and the urine creatinine concentration was assessed using the QuantiChrom™ Creatinine Assay Kit from BioAssay Systems (stated in the Materials and Methods Section 4.2). The urinary creatinine concentrations in control mice were higher than the values in the reference range (shown in Table 1) and significantly higher than seen in SCD mice (Table 1, 40.88 ± 20.86 mg/mL in SCD vs. 87.11 ± 17.35 mg/mL in control, *p* = 0.03859). Clearly, low concentrations of urine creatinine reflected a less concentrated urine in SCD mice. There were no differences in the creatinine amount in the urine collected for 24 h in SCD and control mice (Table 1, 50.26 ± 13.42 mg in SCD vs. 53.22 ± 11.03 mg in control, *p* = 0.7721). The volume of the urine collected for 24 h in the metabolic cages was shown to be about two-fold larger in SCD mice compared to control mice (Table 1, 1.38 ± 0.34 mL in SCD vs. 0.62 ± 0.11 mL in control, *p* = 0.0168), reflecting the urine concentration defect in SCD mice. The levels of urinary Fe, normalized to the urinary creatinine (Table 1, 3.96 ± 2.27 pg/mg in SCD vs. 1.93 ± 0.68 pg/mg in control, *p* = 0.1906), and the amount of Fe released in the urine for 24 h (Table 1, 131.55 ± 48.36 pg in SCD vs. 99.07 ± 39.99 pg in control, *p* = 0.4141) were slightly increased in SCD mice, but these differences did not reach the level of statistical significance.

We assessed kidney function by measuring glomerular filtration rate (GFR) in three male and three female SCD mice by transdermal measurement of sinistrin-Fluorescein Isothiocyanate (sinistrin-FITC) with a small fluorescent detector (MediBecon GmbH, Mannheim, Germany) that recorded the intensity of circulating sinistrin-FITC (Appendix A, raw data, three male and three female SCD mice and control mice, the Materials and Methods Section 4.3). No statistically significant differences in GFR between male and female mice were found (Appendix A). Combined GFR results from six mice are shown in Table 1. As the reference ranges for murine GFR vary significantly depending of strain, age and method of detection [31], we could not find the ranges corresponding to the mice used in this study. The GFR levels were significantly lower in SCD mice compared to control mice (Table 1, 91.1 ± 2.5 µL/min in SCD mice vs. 146.4 ± 30.4 µL/min in control mice, *p* = 0.0067). The reduced GFR levels indicated impaired renal function in SCD mice.

To assess whether SCD mice demonstrate Fe accumulation in organs similar to those of SCD patients, we performed Perl’s Prussian blue staining of sections from paraffin-embedded organs of 4-month-old SCD and control mice (N = 6, three males and three females per group) (Figure 1).The sections were stained as described in Materials and Methods Section 4.4. Images were taken using an Olympus 1 × 51 microscope with an Olympus DP 72 camera and were examined with 100× and 400× magnification (Figure 1). We did not find Fe accumulation in the heart and lung of SCD mice (Figure 1A, representative images of organ sections). However, we observed non-granular, pale blue, cytoplasmic ferritin-bound Fe staining in the SCD hepatocytes (Figure 1B). In contrast, significant Fe accumulation was found in the liver Kupfer cells (Figure 1B, arrowhead) and in the cytoplasm of renal proximal tubular cells (Figure 1C). We observed coarse, dark blue particles, corresponding to hemosiderin staining, in renal epithelial cells. Fe accumulation was not found in the renal medulla of SCD mice (Figure 1C). Fe accumulation patterns were similar in male and female SCD mice. Control mice did not accumulate Fe in the heart, lung, liver, and kidney (Figure 1A–C).

Thus, despite the higher Fe levels in the circulation, only epithelial cells of the renal cortex and Kupffer cells of the liver accumulated significant amounts of Fe in SCD mice.

### 2.2. Iron Intake and Storage in Renal Cortex

Fe accumulation in the renal proximal tubules may be caused by increased Fe intake into the epithelial cells of proximal tubules. Increased levels of cellular intake may be associated with an increased level of urinary Fe, higher TSAT or increased expression levels of TfR1 and DMT1. Urinary Fe and TSAT were not significantly higher in SCD mice compared to control mice (Table 1), thus we assessed expression of TfR1 and DMT1 in the renal cortex. The expression of TfR1 and DMT1 in SCD and control mice was analyzed in the renal cortex by real-time RT-PCR and Western blot (WB) as described in Materials and Methods Section 4.5 and Section 4.6. The levels of *TfR1* mRNA were significantly increased in SCD mice (Figure 2A, *TfR1-Actb* delta Ct, −4.26 ± 0.14 in SCD vs. −4.69 ± 0.12 in control, *p* = 0.0421), but the levels of mRNA expression of the *Slc11a2* gene that encodes DMT1 were similar in SCD and control mice (Figure 2B, *Slc11a2-Actb* delta Ct, −1.68 ± 0.26 in SCD vs. 1.71 ± 0.11 in control, *p* = 0.9236). The levels of TfR1 protein evaluated by WB were also elevated in SCD mice, compared to control mice (Figure 2 C,D, 21.08 ± 0.26 units in SCD vs. 0.32 ± 0.71, *p* = 0.0029).

Ferritin is a protein responsible for storing Fe primarily in hepatocytes. In contrast, renal epithelial cells recycle Fe and prevent Fe loss in the urine. Despite the large number of mitochondria in the epithelial cells of renal proximal tubules and the high demand for intracellular Fe, the tubular epithelial cells do not store Fe and return it to circulation. The levels of ferritin heavy chain (*Fth1*) mRNA were higher in SCD mice compared to control mice (Figure 2E, *Fth1-Actb* delta Ct 5.34 ± 0.62 in SCD vs. 4.86 ± 0.41 in control, *p* = 0.0171). The levels of FtH protein measured in renal cortex homogenate by ELISA were also significantly elevated (Figure 2F, 95.64 ± 9.04 in SCD vs. 28.93 ± 4.32 in control, *p* = 1.08 × 10^−5^). The WB demonstrated significantly increased expression of FtH in SCD mice (Figure 2G,H, 1.36 ± 0.17 units in SCD vs. 0.15 ± 0.05, *p* = 2.8 × 10^−5^).

Thus, higher Fe intake in the renal cortex may be associated with higher TfR1 expression. FtH levels in the renal cortex of SCD mice were significantly higher, indicating cellular Fe accumulation.

### 2.3. Iron Export from Epithelial Cells

FPN is the only known exporter of Fe and is most abundant in the renal cortical region within proximal convoluted tubules [32]. The mRNA levels for the *Slc40a1* gene encoding FPN assessed by real-time RT-PCR were compatible in SCD and control mice (Figure 3A, *Slc40a1-A* delta Ct, −1.24 ± 0.39 in SCD vs. −1.9 ± 0.06 in control, *p* = 0.1165). The protein levels of FPN were slightly reduced in SCD mice (Figure 3B,C, FPN/β-actin, 0.87 ± 0.07 units in SCD vs. 1.09 ± 0.11 units in control, *p* = 0.1295), but the differences were not statistically significant.

Two extracellular ferroxidases, CP and Heph, oxidize Fe(II) (ferrous Fe) into Fe(III) (ferric Fe), facilitating its release from FPN and binding to Tf in circulation. Reduced levels of ferroxidases may reduce the rate of Fe export and increase intracellular Fe levels. CP is mainly synthesized in the liver and released into the circulation, but it is also expressed in the kidney [33]. The expression of *CP* and *Heph* in the kidney was assessed by real-time RT-PCR, as described in Materials and Methods Section 4.5. We observed low levels of *CP* mRNA expression in the renal cortex, which were slightly higher in SCD mice compared to control mice (Figure 3D, *CP-Actb* delta Ct, −8.314 ± 0.35 in SCD vs. −9.036 ± 0.12 in control, *p* = 0.0728). Analysis of CP protein levels in the renal cortex homogenates by WB also demonstrated slightly increased levels in SCD mice (Figure 3E,F, CP/β-actin 2.25 ± 0.54 units in SCD vs. 1.00 ± 0.24 units in control, *p* = 0.0787). *Heph* mRNA is expressed in the small intestine, colon, spleen, placenta, and kidney [34]. We observed *Heph* expression in the renal cortex of SCD and control mice by real-time RT-PCR, but there was no difference in the expression levels between SCD and control mice (Appendix A, *Heph-Actb* delta Ct, −5.059 ± 0.22 in SCD vs. −5.61 ± 0.23 in control, *p* = 0.1067).

Next, we tested FPN expression in the liver. We observed Fe accumulation in the SCD mice hepatocytes (Figure 1B). *Slc40a* mRNA levels were measured by real-time RT-PCR and normalized to the levels of ribosomal protein S13 (*Rps13*) The *Slc40a* mRNA levels were similar in the SCD and control mice (Figure 3G, *Fpn-Rps13* delta Ct, −3.926 ± 0.12 in SCD vs. −3.817 ± 019 in control, *p* = 0.6442). In contrast, protein levels of FPN in the liver, tested by WB, were significantly higher in SCD mice than in control mice, suggesting reduced levels of plasma HAMP in SCD (Figure 3H,I, FPN/β-actin 1.09 ± 0.08 in SCD vs. 0.49 ± 0.07 in control, *p* = 0016).

Thus, despite a significant Fe accumulation in the renal proximal epithelial cells, levels of FPN, CP, and Heph were not significantly changed in SCD mice.

### 2.4. Increased HAMP Expression in the Renal Cortex of SCD Mice

In SCD mice, we observed a significant increase in hepatic FPN; however, renal FPN levels did not show a similar increase. FPN levels are negatively regulated by HAMP, which is predominantly produced in the liver and released into the circulation. HAMP can enter the kidney from the circulation by plasma filtration in glomeruli or be produced by tubular cells [35]. We measured HAMP levels in the serum using the HAMP-Murine Compete^TM^ ELISA kit (Intrinsic Life Sciences (La Jolla, CA, USA), Materials and Methods Section 4.6), and observed significantly lower levels of HAMP in SCD mice compared to control mice (Figure 4A, 0.32 ± 0.05 µM in SCD vs. 0.55 ± 0.04 µM in control, *p* = 0.0081). Since HAMP regulates renal FPN under normal physiological conditions [32], this is an unlikely explanation for the reduced renal FPN levels in SCD mice. To evaluate the expression of HAMP in the renal cortex, we performed real-time RT-PCR and found higher levels of *HAMP* mRNA in SCD compared to control mice (Figure 4B, *HAMP-Actb* delta Ct, −13.41 ± 0.58 for SCD vs. −15.29 ± 0.41 for control, *p* = 0.0189). HAMP protein levels in the renal cortex homogenates detected by ELISA were significantly increased in SCD compared to control mice (Figure 4C, 148.80 ± 30.41 ng/mg protein in SCD vs. 8.12 ± 0.01 ng/mg protein in control, *p* = 0.0001).

Immunostaining of renal sections with rabbit anti-HAMP-25 antibodies (Materials and Methods Section 4.4) demonstrated the presence of cytoplasmic HAMP in the distal tubules (Figure 4D, brown color). Significantly higher levels of expression in SCD mice compared to control mice were measured by ImageJ (Figure 4E, 0.71± 0.01 OD units in SCD vs. 0.63 ± 0.01 OD units in control, *p* = 3.5 × 10^−8^). Double immunostaining for Fe and HAMP showed that cells expressing HAMP did not accumulate Fe (Figure 4F, blue color—Fe, brown color—HAMP). In contrast, Fe accumulation was found mostly in proximal tubules located next to HAMP-expressing distal tubules.

HAMP expression is activated by inflammation and Fe overload, while anemia and hypoxia inhibit it. HAMP regulation in response to Fe is controlled by bone morphogenic protein 6 (BMP6) [36]. Expression of BMP6 mRNA was detected in the renal cortex, but no significant differences were observed between SCD and control mice (Figure 5A, *Bmp6-Actb* delta Ct, −2.45 ± 0.19 in SCD vs. −2.48 ± 0.09 in control, *p* = 0.8977).

Next, we evaluated kidney inflammation in SCD mice. The mRNA expression levels of the pro-inflammatory cytokines IL-1β (Figure 5B, *IL1B-Actb* delta Ct, −3.98 ± 0.39 in SCD vs. −11.19 ± 0.72 in control, *p* = 6.7 × 10^−5^) and IL-6 (Figure 5C, *IL6-Actb* delta Ct, −6.03 ± 0.79 in SCD vs. −11.10 ± 1.06 in control, *p* = 0.0084) were significantly higher in SCD mice compared to control mice. Using immunostaining of renal sections with the specific rat-anti-mouse macrophage antibody (F4/80, Materials and Methods Section 4.4), we observed increased infiltration of activated macrophages in the cortex (Figure 5D, immunostaining, red color) and medulla (Figure 5E, immunostaining, red color, arrows) in SCD mice. Activation of the endothelial cells of the renal capillaries was demonstrated by immunostaining of intracellular adhesion molecule 1 (ICAM1) with rat—anti-mouse ICAM antibody (Figure 5F, ICAM1, red color). Together, these results demonstrated activation of renal inflammation that may stimulate HAMP production.

## 3. Discussion

We have shown here that SCD mice accumulate Fe in the renal proximal tubules in the form of hemosiderin. The levels of FPN in the liver were increased, but unaffected in the kidneys. Chronic hemolysis in SCD mice may lead to systemic Fe overload and Fe accumulation in the organs. Chronic hemolysis in SCD mice was evidenced by the low hematocrit, reduced RBC counts, decreased hemoglobin, and higher reticulocyte counts (Table 1). We also observed increased serum Fe levels in SCD mice. Surprisingly, TIBC levels were also higher in SCD mice. The higher TIBC is usually associated with Fe deficiency; however, it could be due to inflammation, liver disease, or nephrotic syndrome [37]. Fe deficiency is not generally recognized as an issue in SCD patients, as it is thought that the increased gastrointestinal absorption of Fe, and hemolysis causing Fe release from RBC in the circulation, provide a sufficient source of Fe. A large cross-sectional study demonstrated that most SCD patients had normal TIBC. However, about 10% of them had high TIBC, indicating Fe-deficient anemia [38,39]. In mice, gastrointestinal Fe absorption is limited by dietary Fe, which cannot be significantly increased without changing the diet. Fe deficiency and reduced TSAT in SCD mice may be attributed to increased urinary Fe loss and the accumulation of Fe in the liver and kidney. Although higher serum Fe levels were observed in SCD mice, serum Fe is not a reliable indicator of Fe overload [40].

Pearl’s Prussian blue staining showed a pale blue cytoplasmic color in hepatocytes associated with ferritin accumulation and dark blue particles or granules of hemosiderin in the renal proximal tubular epithelial cells of SCD mice (Figure 1B,C). The differences in Fe accumulation between the liver and kidney suggested that organ-specific factors influence Fe metabolism. The normal kidney does not accumulate Fe, but glomerular renal diseases are associated with Fe accumulation in proximal tubules [41]. It was suggested that Fe reabsorption is increased when there is a high amount of TfR1-bound Fe leaking from the injured glomeruli [41]. Glomerulopathy is common in SCD patients, and it is also observed in SCD mice [42,43].

Consequently, we evaluated the expression of proteins involved in renal Fe recycling (Figure 2). Alterations of TfR1 and DMT1 expression are one of the ways to control intracellular Fe uptake. Both TfR1 and DMT1 levels are regulated post-transcriptionally via Fe-induced dissociation of IRPs from Fe-responsive elements (IREs) in the 3′-UTR (untranslated region), resulting in destabilization of *TfR1* and *Slc11a2* mRNA [44,45]. The TfR1 expression in various tissues is regulated by organ-specific cellular Fe requirements. In most non-erythroid cells, elevated intracellular Fe reduces the stability of mRNA and protein levels of TfR1 [4]. In contrast, TfR1 expression in erythroid cells is independent of the cellular Fe levels [45]. Surprisingly, despite the significant accumulation of intracellular Fe, we observed increased TfR1 levels in the renal cortex of SCD mice. Interestingly, in the streptozotocin-induced diabetic rat, renal Fe accumulation is also associated with increased renal TfR1 expression [46]. Therefore, the kidney may respond to the higher levels of Tf-bound Fe filtered through the glomeruli by increasing TfR1 expression via an unknown mechanism. Moreover, TfR1 expression is not consistent throughout the cells of proximal tubules, being highest in the early proximal tubule, and gradually decreasing along the length of the tubule [47]. Thus, the largest amount of Fe, filtrated through the glomeruli, is absorbed in the early proximal tubules. Further study is required to determine how TfR1 expression is regulated in renal proximal cells.

In contrast to the elevated *TfR1* expression, levels of *Slc11a2* expression were the same in SCD and control mice. Scl11a2 expression was significantly higher than that of TfR1 (compare Figure 1A,B). Thus, we speculate that epithelial cells may have a greater capacity for handling intracellular Fe due to elevated Slc11a2 mRNA expression or *Slc11a2* mRNA being degraded at higher intracellular Fe levels. Furthermore, the pattern of intracellular Fe staining in the epithelial cells resembled that of hemosiderin-bound Fe, which is less available for Fe metabolism [48].

As expected, ferritin levels were significantly higher in SCD mice, indicating that the renal epithelial cells had a high Fe content (Figure 2E–H). High ferritin levels are unusual for renal epithelial cells, which are involved in Fe recycling instead of storage, and thus return it to the circulation. Fe-dependent dissociation of IRPs from the 5′-UTR of ferritin mRNA regulates ferritin expression by releasing the translation block [4].

The low availability of hemosiderin-bound Fe for binding to IRPs may also explain why renal FPN mRNA levels remained similar in SCD and control mice, despite the Fe accumulation in the epithelial cells of the proximal tubules of SCD mice. Another possible explanation for unaltered FPN levels is increased HAMP levels. Previous studies have reported lower [49], equal [18] or even higher [50] serum HAMP concentrations in SCD patients, which can be explained by the heterogeneity of the studied groups. Our results demonstrated reduced levels of serum HAMP (Figure 4A) and increased liver FPN levels in SCD mice (Figure 3G–I). Low serum HAMP cannot explain why renal FPN is not upregulated in SCD mice. Although HAMP is expressed in the kidney, the regulation and function of renal HAMP are not completely understood. Under normal Fe intake, the renal HAMP/FPN axis is not necessary for maintaining systemic Fe homeostasis, and it is more important for regulating renal Fe levels [32]. The role of HAMP in regulating renal Fe metabolism has been demonstrated in HAMP knockout mice [35]; the kidneys of these mice showed a decrease in TfR1 and an increase in ferritin and FPN expressions. Renal HAMP is colocalized with calbindin, a marker of distal convoluted tubules and cortical collecting duct [32]. Our results also demonstrated increased levels of HAMP in the renal distal tubules (Figure 4D,F). Interestingly, HAMP is expressed in the distal tubules, while Fe accumulates in the proximal tubules (Figure 4F). Therefore, our results do not support autocrine regulation of renal FPN by renal HAMP, though HAMP could be involved in paracrine regulation. Indeed, FPN is expressed on the basolateral membrane of the proximal epithelia tubules, while HAMP may be secreted through the basolateral membrane of the adjusted distal tubules. In contrast to the SCD mice, in the mouse model of phenyl hydrazine-induced anemia, HAMP expression was decreased in renal distal tubules [51]. Thus, anemia is not a factor influencing renal HAMP expression. HAMP regulation in response to Fe is controlled by BMP6 signaling [52]. We did not find significant differences in BMP6 expression in the kidneys of SCD and control mice (Figure 5A). Additionally, the expression of the *HAMP* gene in the kidney is increased in wild-type mice fed with a high-Fe diet but remains unaltered in hemochromatosis mice [32]. Our observations and these results do not support the idea that renal *HAMP* expression is regulated locally by renal Fe. Renal inflammation may stimulate renal HAMP expression independently of BMP6 [53]. In SCD mice, continuous intravascular hemolysis releases free heme and hemoglobin into the circulation, causing endothelial cell inflammation and macrophage activation [54]. We found a significant increase in vascular inflammation in SCD mice, evidenced by an increase in ICAM1 expression (Figure 5F). We also found an increase in activated macrophage infiltration in the cortex and medulla of SCD mice (Figure 5D,E). Both activated endothelial cells and macrophages produce pro-inflammatory cytokine IL-1β, and, additionally, macrophages produce IL-6. IL-1β and IL-6 induce HAMP expression via the STAT signaling pathway [55,56]. Indeed, our data demonstrated increased mRNA levels of IL-1β and IL-6 in the kidney cortex of SCD mice (Figure 5B,C). In SCD patients, kidney Fe accumulation correlates with the severity of hemolysis but not with blood transfusion rates [13]. Thus, hemolysis products may play a major role in renal Fe accumulation, inducing renal inflammation and the expression of renal HAMP.

## 4. Materials and Methods

### 4.1. Mice

All mice were maintained in a pathogen-free environment, and all animal studies were approved by the Institutional Animal Care and Use Committee (IACUC) at Howard University. The Townes mice, here referred to as SCD mice, were obtained from the Jackson Laboratory (stock number-013071, B6; 129-Hbb^tm2(HBG1,HBB*)Tow/^Hbb^tm3(HBG1,HBB)Tow^ Hba^tm1(HBA)Tow/^J, Bar Harbor, ME, USA) and bred in the Howard University animal facility. The SCD mice lack mouse α- and β-globin genes and carry two copies of the human α1-globin gene and two copies of the human γ-globin and β^S^-globin genes. SCD Townes mice do not express mouse hemoglobin and have approximately 94% human sickle hemoglobin and 2–6% of human fetal hemoglobin [57]. Control mice do not express mouse hemoglobin but express 100% of normal human hemoglobin A and have hα1 and β^A^ genes. Genotypes for all animals were confirmed by PCR with primers: forward—5′-GATATATCTTAGAGGAGGGC-3′ and reverse—5′-CCAACTTCATCCACGTTCAC-3′ followed by Bsu36I digestion (New England Biolab, Ipswich, MA, USA). The restriction digestion was used to distinguish β^S^ and β^A^ alleles. β^A^ fragments are digested, but β^S^ fragments are resistant to Bsu36I digestion. SCD mice replicate hematologic and organ pathologies observed in SCD patients [58,59]. Male and female mice 4–6 months of age were used for all experiments. Body weight at 6 months of age was 29.7 ± 1.1 g for control males, 25.5 ± 1.6 g for control females, 30.4 ± 1.5 g for SCD males and 27.5 ± 2.4 g for SCD females. Mice were euthanized by 1.5–2% isoflurane inhalation followed by cervical dislocation, and organs were collected and either fixed in formalin or stored fresh-frozen at −80 °C. Both male and female mice were used for the study.

### 4.2. Hematology, Plasma Biochemistry and Iron Quantification

The automated veterinary hematology counter Sysmex XN-1000 was used to determine hematological parameters (Sysmex^®^—Roche, Tokyo, Japen). Mouse was anesthetized by isoflurane inhalation, and 200 µL of blood was collected by retro-orbital puncture into an EDTA-coated tube (MTSC-EDTA, Kent Scientific, Torrington, CT, USA). A C57BL/6 chip card was used for results acquisition and analysis. Serum and urine Fe concentrations were assessed by an Fe Assay kit (ab83366, Abcam, Waltham, MA, USA). Briefly, urine was collected using metabolic cages for 24 h. Each mouse was anesthetized by the inhalation of 2% isoflurane, and 200 µL of blood was collected by retro-orbital vein puncture. For serum preparation, blood samples were incubated at room temperature for 30 min for clot formation. The clot was removed by centrifugation at 1000× *g* for 10 min. Kit standards and samples (100 µL of each) were mixed with Fe buffer and Fe probe, and incubated at 37 °C for 60 min. Optical density (OD) was measured at 590 nm using a microplate reader (iMark, Bio-Rad). A standard curve was used for calculation of urine concentration. TIBC was measured by mouse TIBC ELISA kit (MBS2600978, MyBioSource, Inc., San Diego, CA, USA) according to the manufacturer’s protocol [60]. Briefly, 100 µL of standards and serum samples were added to the wells in antibody-covered strips and incubated at 37 °C for 90 min. After intensive washing, 100 µL of biotinylated antibodies were added to each well and incubated at 37 °C for 60 min. Color reagent was added, and OD was measured at 450 nm at 10 min of incubation. TSAT was calculated using the formula:TCAT = Serum Fe/TIBC × 100, %. 

Urine creatinine was measured by QuantiChrom™ Creatinine Assay Kit (DICT-500, BioAssay Systems) [61]. Briefly, 5 µL of standards and urine samples were added to 200 µL of kit reagent buffer, incubated at room temperature for 20 min, and OD was measured at 492 nm.Medi

### 4.3. GFR Assessment

GFR was assessed using transdermal measurement of sinistrin-fluorescein isothiocyanate (FITC) with a small fluorescent detector (MediBecon GmbH) to detect the intensity of circulating sinistrin-FITC [62]. Mice were anesthetized using isoflurane inhalation, and 100 µL of sinistrin-FITC solution (30 mg/mL, 100 mg/kg of body weight, MediBecon GmbH) was injected through retro-orbital vein plexus. The device was attached to depilated skin on the back of the conscious, freely moving animal for 60 min. The results were analyzed using MB_Studio2 platform (MediBeacon GmbH). Raw data are shown in Appendix A.

### 4.4. Iron Staining and Immunohistochemistry

Four-month-old SCD and control mice (3 males and 3 females per group) were used for microscopic evaluation. Organs isolated from control and SCD mice were fixed in formalin (Fisher Scientific, Waltham, MA, USA) and embedded in paraffin. Four micrometer-thick sections from paraffin-embedded organs were used for Perl’s Prussian blue Fe staining and immunohistostaining. For renal ferric Fe detection (Perl’s Prussian blue), sections were incubated in 5% potassium ferrocyanide solution in 10% hydrochloric acid for 30 min and counterstained with hematoxylin (Fisher Scientific). For immunostaining, heat-mediated antigen retrieval in 10 mM sodium citrate (pH 6.0) buffer was performed. Sections were blocked with 10% goat serum and incubated overnight at 4 °C with rat anti-mouse F4/80 (1:40 dilution, ab16911, Abcam) for macrophage detection, and rat anti-mouse ICAM1 (1:200 dilution, BioLegend) and rabbit anti-HAMP-25 antibodies (1:200 dilution, ab190775, Abcam) for HAMP immunostaining. Sections were incubated with secondary biotinylated anti-rat and anti-rabbit antibody (1:500 dilution, both from Vector laboratories) at room temperature for 1 h, following with incubation with streptavidin-peroxidase conjugate (1:500 dilution, Vector Laboratories, Newark, CA) at room temperature for 20 min. AEC and DAB substrate kits were obtained from Vector Laboratories. Sections were counterstained with hematoxylin (Thermo Fisher Scientific Inc., Waltham, MA, USA). Images were acquired using an Olympus 1 × 51 microscope with Olympus DP 72 camera. Immunostaining of the tissue sections was evaluated blindly by two researchers. Positive staining was quantified using ImageJ Fiji and CellSens Standard (Evident, Olympus Scientific Solutions America, Waltham, MA) software.

### 4.5. RNA Extraction and Real-Time RT-PCR

Total RNA extraction and quantitative real-time PCR were performed as previously described [63]. Briefly, total RNA was extracted from renal cortex using TRIzol (Invitrogen, Thermo Fisher Scientific). Total RNA (100 ng) was reverse transcribed to cDNA using the Superscript kit (Invitrogen, Thermo Fisher Scientific) with hexamers as primers. cDNA was amplified using Roche Light Cycler 480 with SYBR Green1 Master mix (Roche Diagnostics). PCR was carried out with denaturation for 10 s at 95 °C, annealing for 10 s at 60 °C, and extension for 10 s at 72 °C, for 45 cycles. Relative mRNA levels were calculated using the comparative Ct method and normalized to beta-actin mRNA. PCR reactions were performed with gene-specific primers that spanned at least one intron. Primers for RT-PCR are shown in Table 2.

### 4.6. Western Blot

Renal cortex sections were cut from frozen kidneys (1–3 mg) and were homogenized in modified RIPA buffer (10 mM Tris-HCl, pH 8.0, 0.15 M NaCl, 1% NP40, 0.1% SDS) supplemented with protease inhibitor cocktail (Sigma-Aldrich, Burlington, MS, USA). The protein concentration was measured by using the detergent compatible (DC) protein assay kit (Bio-Rad). Samples (30 mg of protein per lane) were resolved by SDS-PAGE on 10% Bis-Tris gels (Invitrogen), transferred to polyvinylidene difluoride (PVDF) membranes, and immunoblotted with appropriate antibodies (1:1000 dilution), following with secondary anti-rabbit-peroxidase or anti-mouse-peroxidase conjugates (1:5000 dilution, Sigma-Aldrich). Millipore Sigma Immobilon Western chemiluminescent HRP substrate (Sigma-Aldrich) was used for result visualization. The ChemiDoc MP Imaging system with Image Lab software (Bio-Rad) was used to acquire and quantify WB images, which were then normalized to beta-actin. The list of antibodies for WB is shown in Table 3. Full-sized WB membranes are shown in Appendix A.

### 4.7. Hepcidin and Ferritin Elisa

HAMP levels in the plasma and kidney homogenate were assessed using HAMP-Murine Compete^TM^ ELISA kit (HMC-001, Intrinsic Life Sciences) [64]. HAMP protein concentrations in renal cortex homogenate were normalized to the total protein concentration in homogenate. Plasma ferritin levels were assessed using mouse ferritin ELISA kit (ab157713, Abcam), according to the manufacturer’s instructions. Briefly, standards and 40-fold diluted plasma (100 µL) were loaded to the wells in the assay plate, incubated for 60 min at room temperature, intensively washed and incubated with enzyme-antibody conjugate for 10 min. TMB substrate solution (100 µL per well) was added, and the plate was incubated for 10 min before stop solution (100 µL per well) was added. Then, OD was measured at 450 nm using a microplate reader (iMark, Bio-Rad).

### 4.8. Statistical Analysis

Statistical analysis was performed using Prism 6.0 software (GraphPad, San Diego, CA, USA). Data were analyzed using 2-way ANOVA with the Tukey post hoc test for 4 groups or unpaired parametric Student’s *t*-test for 2 groups. Results were expressed as mean ± standard deviation (STD). Differences between groups were considered significant if *p* < 0.05.

## 5. Conclusions

Our findings provide evidence that renal inflammation may induce upregulation of HAMP expression in the renal cortex of SCD mice, leading to Fe accumulation due to higher serum Fe levels and elevated TfR1 levels without significant changes in FPN levels. Whether HAMP is upregulated in the renal cortex of SCD patients is still unknown. Our results serve as a foundation for further research into the renal Fe accumulation in SCD and other forms of chronic kidney disease (CKD). Elucidating the molecular mechanisms of renal Fe accumulation may lead to the development of novel approaches to reduce renal Fe and slow the progression of CKD.

## Figures and Tables

**Figure 1 ijms-24-10806-f001:**
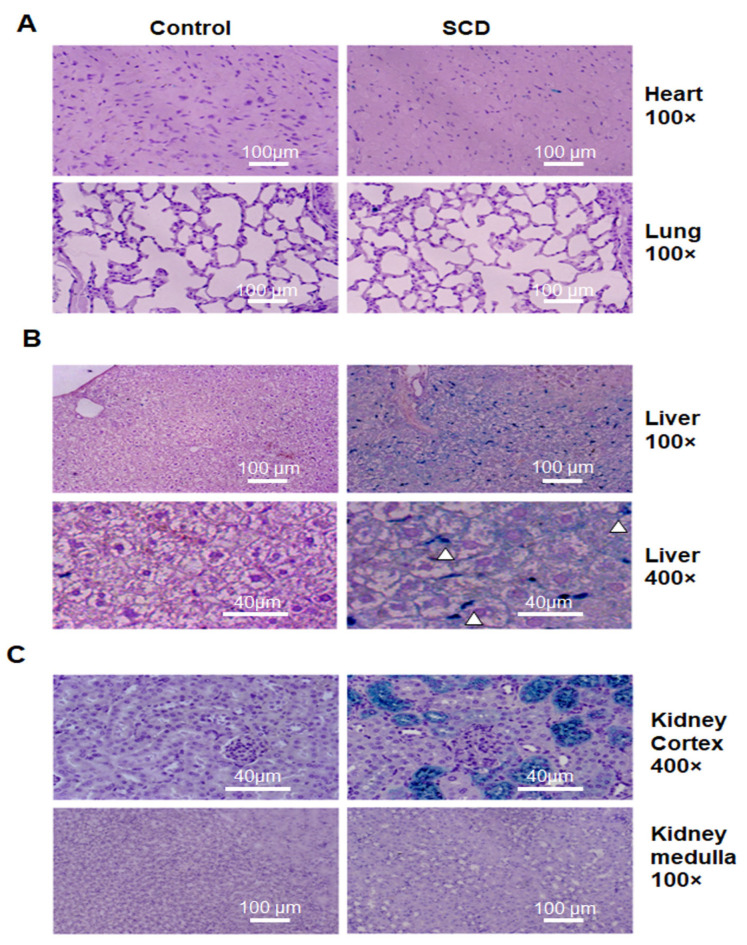
Fe accumulation in renal proximal tubules of SCD mice. Representative images of Perl’s Prussian blue and hematoxylin staining from control and SCD mice are shown (N = 6). (**A**) Fe accumulation was not observed in the heart and lungs of SCD or control mice. Original magnification 100×. (**B**) Fe accumulation in the hepatocytes (light blue color) and Kupfer cells (dark blue color and arrowheads) in SCD mice. Original magnification 100× and 400×. (**C**) Fe accumulation in the renal epithelial cells of proximal tubules (dark blue color and arrows) in the renal cortex of SCD mice. Original magnification 400×. No Fe accumulation was observed in the medulla of either SCD or control mice. Original magnification 100×. Scale bars (100 µm for 100× magnification and 40 µm for 400× magnification) are shown. Hematoxylin was used for counterstaining.

**Figure 2 ijms-24-10806-f002:**
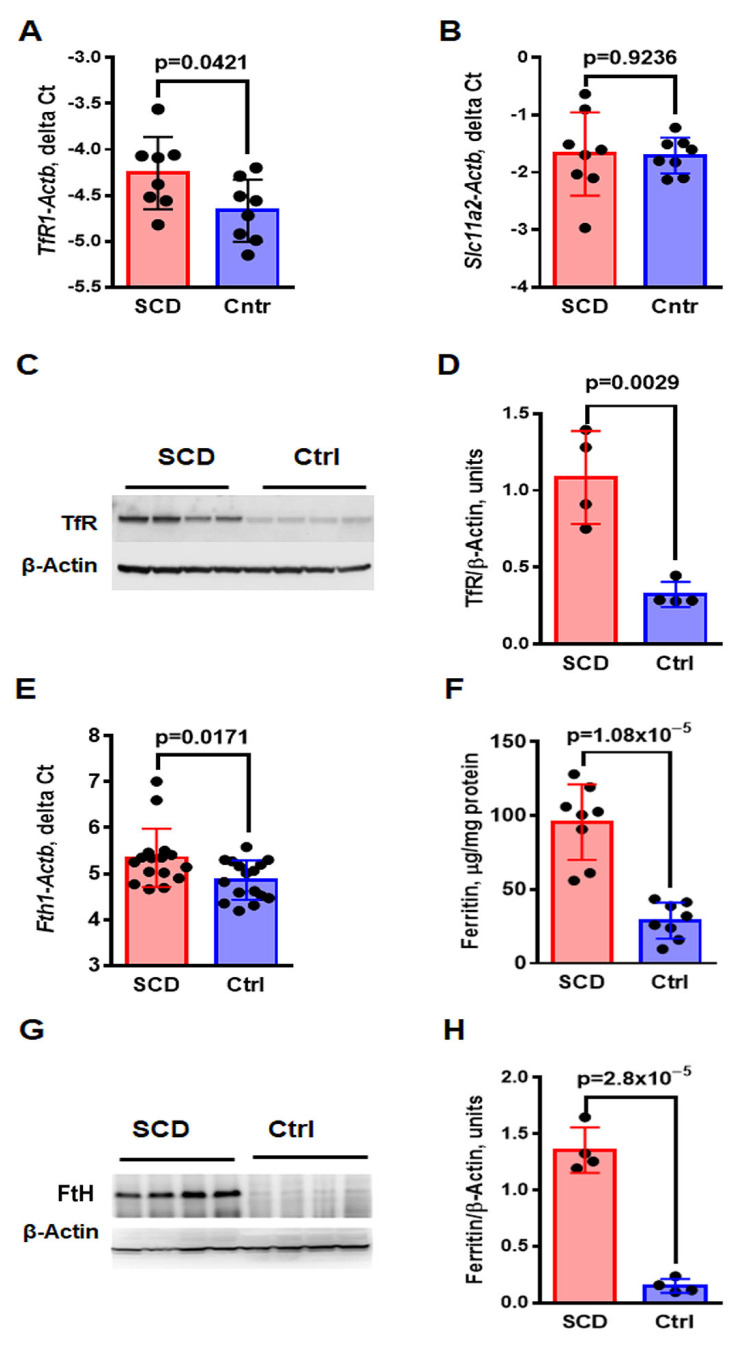
Increased expression of TfR1 and ferritin in the renal cortex of SCD mice. (**A**,**B**) Levels of mRNA expression of *TfR1* (**A**) and DMT1 (*Slc11a2*, (**B**)) in the renal cortex of SCD and control mice were determined by real-time RT-PCR. (**C**,**D**) TfR1 protein expression was assessed by WB (**C**). Quantification of TfR1 WB results (**D**). (**E**,**F**) Levels of mRNA (**E**) and ELISA (**F**) of ferritin heavy chain 1 (Fth1) expression. (**G**,**H**) WB (**G**) and quantification (**H**) of Fth1 expression. For assessment of mRNA levels, real-time RT-PCR of mRNA isolated from the renal cortex of SCD and control mice (N = 8) was performed. Results are normalized to β-actin mRNA levels (*Actb*) and shown as delta Ct. For WB, tissue homogenates (30 mg) were resolved by PAGE-SDS and transferred to the PVDF membrane. The membrane was probed with rabbit anti-TfR1 (Abcam) (**C**), rabbit anti-Fth1 (SCBT) (G) and mouse anti-β-actin (SSCBT) (**C** and **G**) antibodies. WB quantification was performed using Image Lab software (Bio-Rad, Hercules, CA, USA), with β-actin as a loading control. Means and STD are shown in the graphs. Each dot represents a value obtained from an individual mouse. *p* < 0.05 is considered significant.

**Figure 3 ijms-24-10806-f003:**
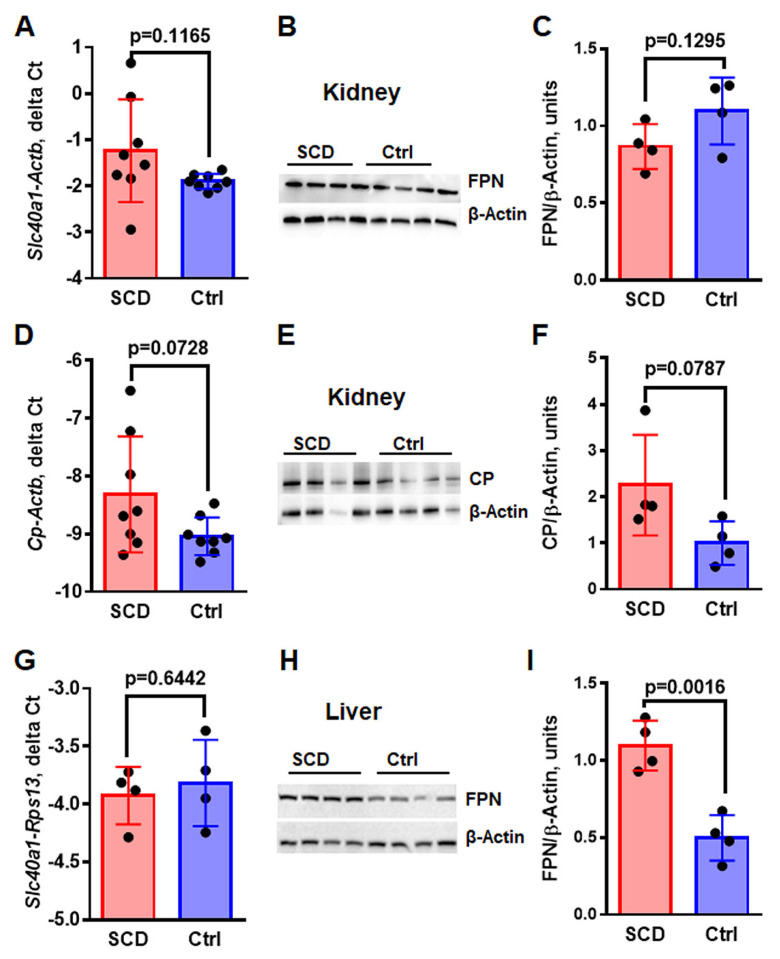
FPN protein levels were increased in the liver, but not in the kidney of SCD mice. (**A**–**C**) Renal FPN mRNA levels (*Slc40a1*, (**A**)), WB (FPN, (**B**)) and WB quantification (**C**). (**D**–**F**) Renal CP mRNA levels (*CP*, (**D**)), protein levels assessed by WB (CP, (**E**)) and WB quantification (**F**). (**G**–**I**) Hepatic FPN mRNA levels (*Slc40a1*, (**G**)), protein levels assessed by WB (FPN, (**H**)) and WB quantification (**I**). For assessment of mRNA levels, real-time RT-PCR of mRNA isolated from the renal cortex and liver of SCD and control mice (N = 8) was performed. Results are normalized to β-actin mRNA levels (*Actb*) and shown as delta Ct. For WB, tissue homogenates (30 mg) were resolved by PAGE-SDS and transferred to the PVDF membrane. The membrane was probed with rabbit anti-FPN (Invitrogen, Waltham, MA, USA) (**B**,**H**), rabbit anti-CP (Abcam) (**E**) and mouse anti-β-actin (SSCBT) (**B**,**E**,**H**) antibodies. WB quantification was performed using Image Lab software (Bio-Rad). Means and STD are shown in the graphs. Each dot represents a value obtained from an individual mouse. *p* < 0.05 is considered significant.

**Figure 4 ijms-24-10806-f004:**
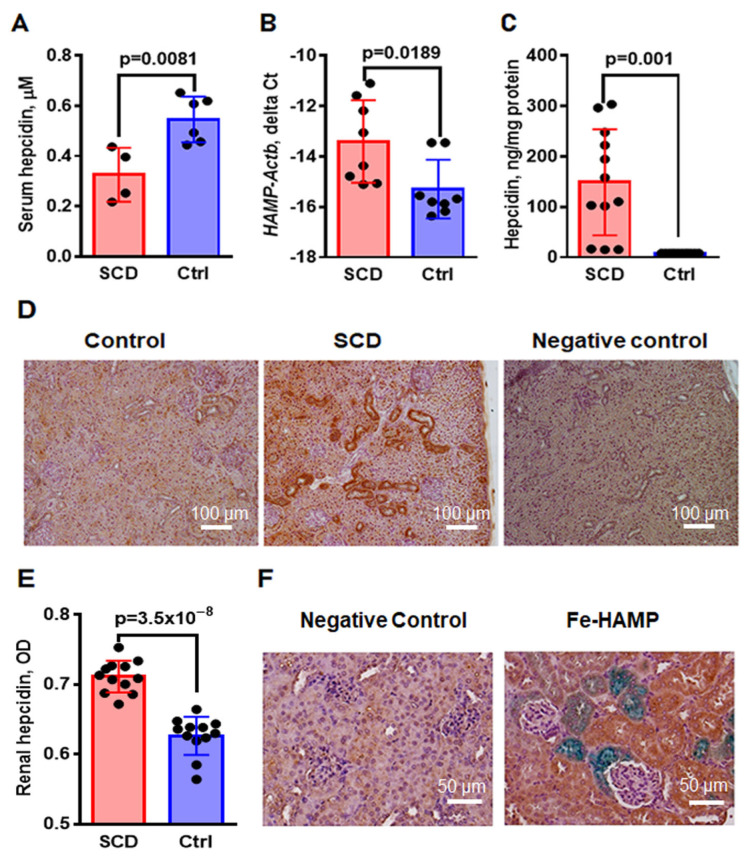
Upregulation of HAMP expression in the renal cortex of SCD mice. (**A**) Serum levels of HAMP detected by ELISA. (**B**) Renal *HAMP* mRNA levels in SCD and control mice. For assessment of mRNA levels, real-time RT-PCR of mRNA isolated from the renal cortex of SCD and control mice was performed (N = 8). Results were normalized to β-actin mRNA levels (*Actb*) and shown as delta Ct. (**C**) HAMP protein concentrations in renal cortex homogenate measured by ELISA and normalized to the total protein concentration in the homogenate. HAMP levels in the plasma (**A**) and kidney homogenate (**C**) were assessed using the HAMP-Murine Complete ELISA kit (Intrinsic Life Sciences). (**D**) Representative images of HAMP immunostaining (brown color) in the kidney of control and SCD mice are shown (N = 6). Rabbit anti-HAMP 25 antibodies (Abcam) were used for immunostaining. Section of SCD mouse kidney stained with non-specific rabbit primary antibodies was used as a negative control. Hematoxylin was used for counterstaining. Original magnification 100×. Scale bars (100 µm for 100× magnification) are shown. (**E**) Quantification of positive HAMP immunostaining was performed using ImageJ Fiji software (three random fields in the sections from four mice). (**F**) Representative images of Perl’s Prussian blue staining (blue color) and HAMP immunostaining (brown color) in the kidney of control and SCD mice are shown (N = 6). Hematoxylin was used for counterstaining. Original magnification 200×. Scale bars (50 µm for 200× magnification) are shown. For graphs, means and STD are shown. Each dot represents a value obtained from an individual mouse. *p* < 0.05 is considered significant.

**Figure 5 ijms-24-10806-f005:**
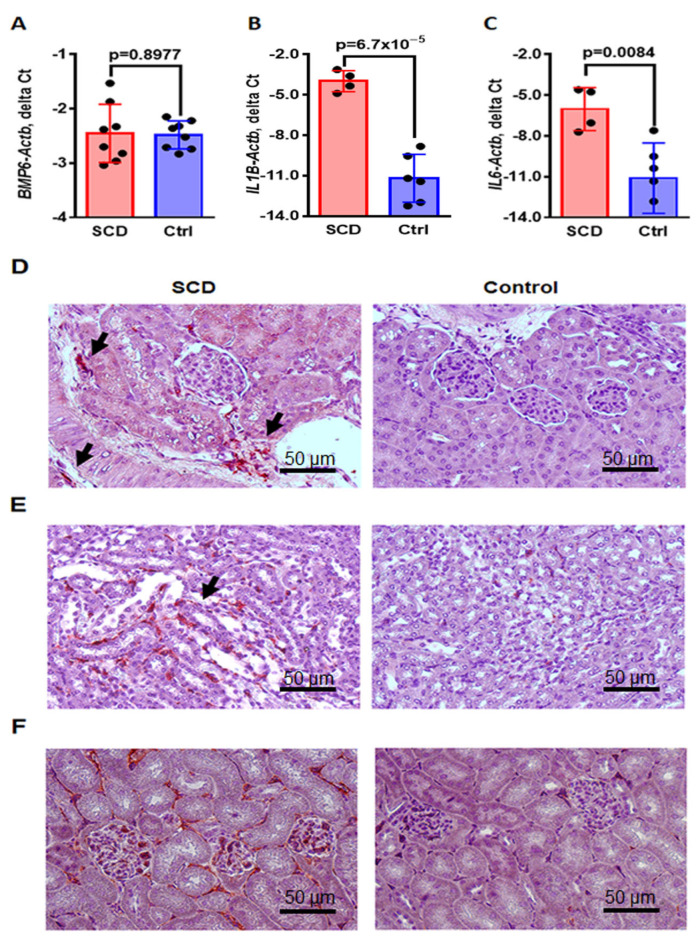
The upregulation of inflammation in the renal cortex of SCD mice. (**A**–**C**) Levels of mRNA expression of *BMP6* (A), IL-1β (*IL1B*, (B)) and IL-6 (*IL6*, (**C**)) in the renal cortex. For assessment of mRNA levels, real-time RT-PCR of mRNA isolated from the renal cortex of SCD and control mice was performed (N = 8). Results were normalized to mRNA β-actin levels (*Actb*)*,* shown as delta Ct. Means and STD are shown in the graphs. Each dot represents an individual mouse. *p* < 0.05 is considered significant. (**D**,**E**) Representative images of immunostaining of macrophages in the renal cortex (**D**) and medulla (**E**) (red color, indicated by arrows) of SCD and control mice (N = 6). Rat anti-mouse macrophage antibody (F4/80, AbDSerotec) was used. Original magnification 200×. Scale bars (50 µm for 200× magnification) are shown. (**F**) Representative images of immunostaining of intracellular adhesion molecule 1 (ICAM1, red color) in the endothelial cells of the renal cortex of SCD and control mice (N = 6). Rat anti-mouse ICAM1 antibody (BioLegend) was used. Original magnification 200×. Scale bars (50 µm for 200× magnification) are shown. Hematoxylin was used for counterstaining.

**Table 1 ijms-24-10806-t001:** Hematological and urinary parameters.

Parameter	SCD	Control	*p*	Reference Range
Hematocrit, %	27.74 ± 1.43	45.02 ± 2.81	1.19 × 10^−5^	35–52 ^8^
Hemoglobin, g/dL	6.4 ± 0.21	11.38 ± 0.95	4.76 × 10^−6^	11.7–17.3 ^8^
RBC ^1^, 10^6^ µL	6.94 ± 0.19	12.65 ± 0.86	6.85 × 10^−7^	7.8–10.6 ^8^
MCV ^2^, fL ^3^	40.02 ± 1.46	33.28 ± 0.6	1.34 × 10^−5^	45–55 ^10^
MCH ^4^, pg	9.33 ±0.12	9.07 ± 0.29	0.2943	15.2–16.2 ^10^
Reticulocytes, %	38.06 ± 2.45	6.35 ± 0.96	1.75 × 10^−9^	3.57–15.2 ^8^
Serum Fe, µg/dL	165.11 ± 38.79	113.81 ± 33.37	0.0618	102.0–190.4 ^8^
TIBC ^5^, µM	532.83 ± 99.71	321.55 ± 149.12	0.05663	250–450 ^9^
TSAT ^6^, %	34.51 ± 9.08	37.81 ± 12.29	0.6814	25–37 ^9^
Urinary creatinine, mg/dL	40.88 ± 20.86	87.11 ± 17.35	0.03859	32.5–63.1 ^10^
Urinary creatinine, mg per 24 h	50.26 ± 13.42	53.22 ± 11.03	0.7721	N/A ^12^
Urine volume per 24 h, mL	1.38 ± 0.34	0.62 ± 0.11	0.0168	0.5–1.3 ^8^
Urine Fe/urine Cr ^7^, pg/mg	3.96 ± 2.27	1.93 ± 0.68	0.1906	0.95–3.6 ^9^
Urine Fe, pg per 24 h	131.55 ± 48.36	99.07 ± 39.99	0.4141	N/A ^12^
GFR ^11^, µL/min	91.1 ± 2.5	146.4 ± 30.4	0.0067	N/A ^12^

^1^ RBC—red blood cells; ^2^ MCV—mean corpuscular volume; ^3^ fL—femtoliters; ^4^ MCH—mean corpuscular hemoglobin; ^5^ TIBC—total Fe binding capacity; ^6^ TSAT—transferrin saturation; ^7^ Cr—creatinine; ^8^ O’Connell et al., 2015 [28]; ^9^ Rodrigues et al., 2014 [29]; ^10^ Mouse phenome database at the Jackson Laboratory (https://phenome.jax.org/strains/7 (accessed on 7 June 2023)); ^11^ Glomerular filtration rate; ^12^ Not available.

**Table 2 ijms-24-10806-t002:** Primers for real-time PCR.

Gene	Forward Primer	Reverse Primer
*Slc40a1*	5′-TTGCAGGAGTCATTGCTGCTA-3′	5′-TGGAGTTCTGCACACCATTGAT-3′
*TfR1*	5′-TCATGAGGGAAATCAATGATCGTA-3′	5′-GCCCCAGAAGATATGTCGGAA-3′
*Dmt1*	5′-GGCTTTCTTATGAGCATTGCCTA-3′	5′-GGAGCACCCAGAGCAGCTTA-3′
*Hamp*	5′-CCATCAACAGATGAGACAGACTAC-3′	5′-TTGCAACAGATACCACACTGG -3′
*Fth1*	5′-TGATGAAGCTGCAGAACCAG-3′	5′-GTGCACACTCCATTGCATTC-3′
*Bmp6*	5′-GCTGAGTTCCGCGTCTACAA-3′	5′-ACCCGGGTGTCCAACAAAAA-3′
*Heph*	5′-TTGTCTCATGAAGAACATTACAGCAC-3′	5′-CATATGGCAATCAAAGCAGAAGA-3′
*CP*	5′-AAAGTCCCTCTGCCTCAGGT-3′	5′-TTTTCCCAGATTGTCCTGGT-3′
*IL6*	5′-TAGTCCTTCCTACCCCAATTT CC-3′	5′-TTGGTCCTTAGCCACTCCTTC-3′
*IL1b*	5′-GACCTTCCAGGATGAGGACA-3′	5′-AGCTCATATGGGTCCGACAG-3′
*Actb*	5′-CCTAGCACCATGAAGATCAAG-3′	5′-AAGGGTGTAAAACGCAGCTC-3′

**Table 3 ijms-24-10806-t003:** Antibodies used for Western blot.

Antibody	Host	WB Dilution	Reference/Supplier
**FPN1/Slc40a1**	Rabbit	1:1000	PA5-22993, Invitrogen/ MTP11-A, Alpha diagnostics
**TfR1**	Rabbit	1:1000	AB84036, Abcam
**CP**	Rabbit	1:1000	Ab48614, Abcam
**FtH**	Rabbit	1:1000	SC-25617, SCBT
**Beta-Actin**	Mouse	1:10000	SC-47778 HRP, SCBT

## Data Availability

All data presented in this data are available upon request from the corresponding author.

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
