# Peer review of "Induction of Hepcidin Expression in the Renal Cortex of Sickle Cell Disease Mice"

_ijms, 2023, doi:10.3390/ijms241310806_

Round 1

Reviewer 1 Report

The paper is interesting and can progress to the next step. The authors should revise the paper according to the following significant comments:

- The abstract section should be more focused on the results of the research. 

- The introduction section is presented very well.

- After the introduction section the authors should present the different methods that they used in order to achieve the results presented in section 2. 

- It is not clear from the paper how the authors get the results. 

- The analysis and the discussion of all the figures presented in the paper should be extended. 

- The authors should connect the methods, the results, and the graph plots presented in the figures. 

- Although the authors present in section 4 the Materials and Methods, it is not absolutely clear the relation to the results of the research. 

- The following papers can be added to the current research: 

1: Vazquez-Meves, G., Kumari, N., Afangbedji, N., Khaibullina, A., Quezado, Z., Nekhai, S., & Jerebtsova, M. (2016). Upregulation of Renal Iron Metabolism in Sickle Cell Disease Mice. In Blood (Vol. 128, Issue 22, pp. 1276–1276). American Society of Hematology. https://doi.org/10.1182/blood.v128.22.1276.1276

2: Nave, Op., & Sigron, M. (2022). A mathematical model for cancer treatment based on combination of anti-angiogenic and immune cell therapies. In Results in Applied Mathematics (Vol. 16, p. 100330). Elsevier BV. https://doi.org/10.1016/j.rinam.2022.100330

Please edit the paper by English mother language

Best regards and good luck

Author Response

Reviewer 1

Q1. The abstract section should be more focused on the results of the research. 

Response: We corrected abstract to be more focused on research.

Q2.  After the introduction section the authors should present the different methods that they used in order to achieve the results presented in section 2. 

Response: The manuscript is written using the journal template and instruction for authors. Instructions are recommended to place the Materials and Methods section after Discussion, not before Results section. We changed the order of Methods to follow the result presentation in the Result section.

Q3. It is not clear from the paper how the authors get the results. 

Response: We significantly corrected the Materials and Methods section and added connections to the Result section. We also added short method description into the figure legends.

Q4. The analysis and the discussion of all the figures presented in the paper should be extended. 

Response:  We extended the analysis and the discussion of all figures as requested.

Q5. The authors should connect the methods, the results, and the graph plots presented in the figures. 

Response: We corrected the methods, the results and the figures, connecting them together.

Q6. Although the authors present in section 4 the Materials and Methods, it is not absolutely clear the relation to the results of the research. 

Response: We corrected Materials and Method section, added references and brief description of all methods.

Q7. The following papers can be added to the current research: 

1: Vazquez-Meves, G., Kumari, N., Afangbedji, N., Khaibullina, A., Quezado, Z., Nekhai, S., & Jerebtsova, M. (2016). Upregulation of Renal Iron Metabolism in Sickle Cell Disease Mice. In Blood (Vol. 128, Issue 22, pp. 1276–1276). American Society of Hematology. https://doi.org/10.1182/blood.v128.22.1276.1276

2: Nave, Op., & Sigron, M. (2022). A mathematical model for cancer treatment based on combination of anti-angiogenic and immune cell therapies. In Results in Applied Mathematics (Vol. 16, p. 100330). Elsevier BV. https://doi.org/10.1016/j.rinam.2022.100330

Response: We respectfully disagree with the reviewer. The first reference is an abstract and not the full paper. The second reference describes cancer cell growth related to various growth factors. It is unclear how it relates to the current study and what it signifies.   

Q8. Please edit the paper by English mother language

Response: We edited English style of the paper.

Reviewer 2 Report

1. Instead of the P column, they should add a column for the biological reference ranges for each analyte because as the results are presented, it is not known whether they are in or out of range.

2. The determination of hemoglobin, hematocrit, and red blood cell count requires biological reference intervals according to altitude above sea level, and this applies to any murine model, so a solid justification for this observation is required.

3. In any case, the glomerular filtration rate would have been determined. Even another renal biomarker will detect early if there is damage to this organ.

4. Calibration bars must be attached to all figures.

5. Age and weight of the mice at the start of the experiments, as well as their gender.

6. With which substance or drug were euthanized rats?.

7. Does the automated equipment have biological reference intervals for mice?.

8. Were the kits used for the different determinations murine-specific?.

Author Response

Q1. Instead of the P column, they should add a column for the biological reference ranges for each analyte because as the results are presented, it is not known whether they are in or out of range.

Response: We added reference ranges for each parameter in Table 1. There are no standard protocols for hematological and urinary parameters in mice, and the results are strongly depending on the protocol and varied between different laboratories. In the Table 1 we combined reference range using results obtained from two reviews and mouse phenome database at the Jackson Laboratory (https://phenome.jax.org/strains/7).

Q2. The determination of hemoglobin, hematocrit, and red blood cell count requires biological reference intervals according to altitude above sea level, and this applies to any murine model, so a solid justification for this observation is required.

Response: We agree with reviewer that hemoglobin, hematocrit, and red blood counts strongly depend on the altitude. The altitude in Washington DC USA, where Howard University is located is 100-125 m above see level. The elevation in Bethesda MD USA, where the National Institute of Health (NIH) is located is 97 m above the see level. We used reference ranges established at NIH. Thus, the elevation is not likely to affect the mouse blood parameters in our study. Moreover, control and SCD mice were housed in the same facility and were exposed to the same oxygen levels.

Q3. In any case, the glomerular filtration rate would have been determined. Even another renal biomarker will detect early if there is damage to this organ.

Response: We agree that glomerular filtration rate (GFR) is an important characteristic of renal function. Glomerular filtration rate (GFR) was assessed using transdermal measurement of sinistrin-fluorescein isothiocyanate (FITC) with a small fluorescent detector (MediBecon GmbH) to detect intensity of circulating sinistrin-FITC. Mice were anesthetized using isoflurane inhalation and 100 µl of sinistrin-FITC solution (30 mg/ml, 100 mg/kg of body weight, Fresenius Kabi Austia GmbH) was injected though retro-orbital vein plexus. The device was attached to a depilated skin on the back of of conscious, freely moving animal for 60 min. The results were analyzed using MB_Studio2 platform (MediBeacon GmbH).

We added description of the procedure to the Materials and Methods sections, added results to Table 1 and added raw results to the Supplemental Materials.

Q4. Calibration bars must be attached to all figures.

Response: We included scale bar to each microphotograph shown in figures 1, 4 and 5.

Q5. Age and weight of the mice at the start of the experiments, as well as their gender.

Response: Male and female mice 4-6 months of age were used for all experiments. Body weight at the 6 months of age was 29.7 ± 1.1 g for control males, 25.5 ± 1.6 g for control females, 30.4 ± 1.5 g for SCD males and 27.5 ± 2.4 g for SCD females.

Q6. With which substance or drug were euthanized rats?.

Response: Mice were euthanized by inhalation of 1.5-2% isoflurane following by cervical dislocation. We added this information in to the Material and Methods section 4.1.

Q7. Does the automated equipment have biological reference intervals for mice?.

Response: Veterinary hematology counter Sysmex XN-1000 was used with C57BL/6 chip card, that is specific for mouse C57BL/6 strain that was used for generation of control and sickle cell disease mice. We included this information in Materials and Methods.

Q8. Were the kits used for the different determinations murine-specific?.

Response:  Total iron binding capacity ELISA kit (MBS2600978, MyBioSource, Inc. CA, USA), mouse ferritin kit (ab157713, Abcam), and Hepcidin-Murine CompeteTM (HMC-001, Intrinsic Life Sciences, CA, USA) were murine-specific. Iron Assay kit (ab83366, Abcam, MA USA) and QuantiChrom™ Creatinine Assay Kit, BioAssay Systems (DICT-500, PA, USA) were not murine-specific, because these chemicals are common in all species. The information about kits was added to Materials and Methods sections.

Reviewer 3 Report

1. Check the abbreviations throughout the manuscript and introduce the abbreviation when the full word appears the first time in the abstract and the remaining for the text and then use only the abbreviation (For example, iron (Fe), red blood cells (RBCs), transferrin saturation (TSAT)  etc.,). Make a word abbreviated in the article that is repeated at least three times in the text, not all words  to be abbreviated.

2. The figure legends should be improved and a proper footnote should be given. All legends should have enough description for a reader to understand the figures without having to refer back to the main text of the manuscript. For example, the necessary abbreviations should be given.

3. The age group (control also) and gender of the animals should be included in the materials and methods under the heading “4.1. Mice”.

4. In the materials and methods, the authors may cite references for standard protocol, instead of mentioning kid or manufacture instructions, if reference is given with it and the same should be added in the reference section.

5. The authors should give the version of the software used in the statistical analysis (material and methods) since it is not mentioned (“4.6. Statistical analysis”).

6. The conclusion seems to be simple. Moreover, the authors may also be included the limitation and future direction of the present findings for a better understanding of the manuscript.

1. The English need improvement since there are some grammatical and syntax errors in the manuscript. For example,

·         in line number 13, the word “Iron” may be as “The iron”;

·         in line number 15, “mouse” as “the mouse”;

·         in line number 20, “kidney” as “the kidney”;

·         in line number 35, “iron in” as “the iron in”;

·         in line number 95, “renal” as “the renal”;

·         in line number 96, “Townes” as “the Townes”;

·         in line number 127, “significant” as “a significant”;

·         in line number 66, “in the circulation” as “to circulation”;

·         in line number 218, “the circulation” as “circulation”;

·         in line number 238, “increase of” as “increase in”;

·         in line number 258, “negative” as “a negative”;

·         in line number 318, “Normal” as “The normal”;

·         in line number 343, “handling of” as “handling”;

·         in line number 350, “the iron” as “iron”;

·         in line number 350, “the circulation” as “circulation”;

·         in line number 360, “normal” as “a normal”;

·         in line number 361, “maintenance” as “the maintenance”;

·         in line number 362 and 363, “regulation” as “the regulation”;

·         in line number 364, “Kidneys” as “The kidneys”;

·         in line number 370, “proximal” as “the proximal”;

·         in line number 375, “expression of Hamp” as “the expression of the Hamp”;

·         in line number 376, “high-iron” as “a high-iron”;

·         in line number 381, “increase of” as “increase in”;

·         in line number 382, “evident by increase ICAM1” as “evidenced by an increase in ICAM1”;

·         in line number 386, “expression” as “of expression”;

·         in line number 389, “the renal” as “renal”;

·         in line number 399, “human” as “the human”;

·         in line number 434 and 467, “renal” as “the renal”;

·         in line number 464, “upregulation” as “the upregulation”.

The grammar mistakes which are not mentioned here are also to be checked and corrected properly.

2. There are some typing mistakes as well, and authors are advised to carefully proof-read the text. For example,

·         in line number 45, the word “export” may be as “exported”;

·         in line number 69, “controls” as “control”;

·         in line number 84, “endocitosis” as “endocytosis”;

·         in line number 185, “primerily” as “primarily”;

·         in line number 186, “recicling” as “recycling”;

·         in line number 299, “organ's” as “organ”;

·         in line number 303, “however it” as “however, it”;

·         in line number 311, “results” as “result”;

·         in line number 325, “Both, TfR” as “Both TfR”;

·         in line number 325, “transcriptionaly” as “transcriptionally”;

·         in line number 345, “hemosiderin- bound” as “hemosiderin-bound”;

·         in line number 370, “accumulats” as “accumulates”;

·         in line number 384, “additionally” as “additionally,”;

·         in line number 435, “tocDNA” as “to cDNA”;

·         in line number 465, “cortes” as “cortex”.

The typos not mentioned here are also to be checked and corrected properly.

Author Response

Q1. Check the abbreviations throughout the manuscript and introduce the abbreviation when the full word appears the first time in the abstract and the remaining for the text and then use only the abbreviation (For example, iron (Fe), red blood cells (RBCs), transferrin saturation (TSAT)  etc.,). Make a word abbreviated in the article that is repeated at least three times in the text, not all words  to be abbreviated.

Response: We checked the abbreviations throughout the manuscript and corrected them according the Reviewer suggestion.

Q2. The figure legends should be improved and a proper footnote should be given. All legends should have enough description for a reader to understand the figures without having to refer back to the main text of the manuscript. For example, the necessary abbreviations should be given.

Response: We corrected figure legends and added a brief explanation of methods.

Q3. The age group (control also) and gender of the animals should be included in the materials and methods under the heading “4.1. Mice”.

Response: We added information about age and sex of mice in the Materials and Methods section under 4.1. For study we used four-to six- months old male and female mice.

Q4. In the materials and methods, the authors may cite references for standard protocol, instead of mentioning kid or manufacture instructions, if reference is given with it and the same should be added in the reference section.

Response: We cited references for the standard protocols and added a brief description of the manufacturer’s protocols.

Q5. The authors should give the version of the software used in the statistical analysis (material and methods) since it is not mentioned (“4.6. Statistical analysis”).

Response: We added the version of the software used in the statistical analysis and re-wrote the section as: “Statistical analysis was performed using Prizm 6.0 software (GraphPad). Data was analyzed using 2-way ANOVA with the Turkey post hoc test for 4 groups or unpaired parametric Student’s t-test for two groups. Results were expressed as mean ± standard deviation (STD). Differences between groups were considered significant if p <0.05.”

Q6. The conclusion seems to be simple. Moreover, the authors may also be included the limitation and future direction of the present findings for a better understanding of the manuscript.

Response: We re-write conclusion, including limitation and future direction.

Comments on the Quality of English Language

Q1. The English need improvement since there are some grammatical and syntax errors in the manuscript. For example,

  • in line number 13, the word “Iron” may be as “The iron”;
  • in line number 15, “mouse” as “the mouse”;
  • in line number 20, “kidney” as “the kidney”;
  • in line number 35, “iron in” as “the iron in”;
  • in line number 95, “renal” as “the renal”;
  • in line number 96, “Townes” as “the Townes”;
  • in line number 127, “significant” as “a significant”;
  • in line number 66, “in the circulation” as “to circulation”;
  • in line number 218, “the circulation” as “circulation”;
  • in line number 238, “increase of” as “increase in”;
  • in line number 258, “negative” as “a negative”;
  • in line number 318, “Normal” as “The normal”;
  • in line number 343, “handling of” as “handling”;
  • in line number 350, “the iron” as “iron”;
  • in line number 350, “the circulation” as “circulation”;
  • in line number 360, “normal” as “a normal”;
  • in line number 361, “maintenance” as “the maintenance”;
  • in line number 362 and 363, “regulation” as “the regulation”;
  • in line number 364, “Kidneys” as “The kidneys”;
  • in line number 370, “proximal” as “the proximal”;
  • in line number 375, “expression of Hamp” as “the expression of the Hamp”;
  • in line number 376, “high-iron” as “a high-iron”;
  • in line number 381, “increase of” as “increase in”;
  • in line number 382, “evident by increase ICAM1” as “evidenced by an increase in ICAM1”;
  • in line number 386, “expression” as “of expression”;
  • in line number 389, “the renal” as “renal”;
  • in line number 399, “human” as “the human”;
  • in line number 434 and 467, “renal” as “the renal”;
  • in line number 464, “upregulation” as “the upregulation”.

The grammar mistakes which are not mentioned here are also to be checked and corrected properly.

Response: We corrected grammatical and syntax errors.

Q2. There are some typing mistakes as well, and authors are advised to carefully proof-read the text. For example,

  • in line number 45, the word “export” may be as “exported”;
  • in line number 69, “controls” as “control”;
  • in line number 84, “endocitosis” as “endocytosis”;
  • in line number 185, “primerily” as “primarily”;
  • in line number 186, “recicling” as “recycling”;
  • in line number 299, “organ's” as “organ”;
  • in line number 303, “however it” as “however, it”;
  • in line number 311, “results” as “result”;
  • in line number 325, “Both, TfR” as “Both TfR”;
  • in line number 325, “transcriptionaly” as “transcriptionally”;
  • in line number 345, “hemosiderin- bound” as “hemosiderin-bound”;
  • in line number 370, “accumulats” as “accumulates”;
  • in line number 384, “additionally” as “additionally,”;
  • in line number 435, “tocDNA” as “to cDNA”;
  • in line number 465, “cortes” as “cortex”.

The typos not mentioned here are also to be checked and corrected properly.

Response: We corrected typos in the manuscript,

Round 2

Reviewer 1 Report

I can confirm that the authors revised the paper according to my major comments. 

Please edit the paper in English editor

Author Response

Q1. Please edit the paper in English editor

Response: We edited paper with English Editor. The corrections are shown in red color.

Reviewer 3 Report

1. The suggestion is not properly carried out and should be rectified. Check the abbreviations throughout the manuscript and introduce the abbreviation when the full word appears the first time in the abstract and the remaining for the text and then use only the abbreviation. For example, in line number 34, the short (Fe) form of iron is not given and similarly for  “transferrin saturation” in line numbers 106 and 424. These types of corrections are need to be carriedout properly for all other abbreviations used in the manuscript.

1. There are some grammatical, alignment and typographical errors are noted in the manuscript and it should be thoroughly checked and corrected throughout the manuscript. For example, in line number 13, the word “kidneys” may be as “the kidneys”; in line number 13, “Iron” as “he iron”; in line number 15, “mouse” as “a mouse”; in line number 17, “transferrin” as “the transferrin”; in line number 23, “observasions” as “observations”; in line number 24, “potentailly” as “potentially”; in line number 26, “upregulation” as “the upregulation”; in line number 40, “that is” as “which is”; in line number 48, “incorporates in” as “incorporates”; in line number 49, “feroportin” as “ferroportin”; in line number 50, “all cells;, and is it consisted” as “all cells, and has or does it consisted”; in line number 80, “hephestin” as “hephaestin”; in line number 108, “transdermal” as “a transdermal”; all over the manuscript, “the Table” as “Table”; in line number 124, “reference” as “the reference”; in line number 163, “describe” as “described”; in line number 164, “Olympus” as “an Olympus”; in line number 187, “expression” as “the expression”; in line number 187, “renal” as “the renal”; in line number 188, “was analyzed” as “were analyzed”; all over the manuscript, “real time” as “real-time”; in line number 201 and 225, “PVDF” as “the PVDF”; in line number 207, “the prevent” as “prevent”; in line number 208, “in the circulation” as “to circulation”; in line number 222, “assessment” as “the assessment”; in line number 232, “but he” as “but the”; in line number 233, “oxidiaze” as “oxidize or oxidized”; in line number 258, “Hepcidin-Murine” as “the Hepcidin-Murine”; in line number 274, “negative” as “a negative”; in line number 284, “presence” as “the presence”; in line number 300, “renal” as “the renal”; in line number 311, “anti mouse” as “anti-mouse”; in line number 315, “ware” as “were”; in line number 345, “unknown” as “an unknown”; in line number 360, “explanation of” as “explanation for”; in line number 365, “normal” as “a normal”; in line number 385, “additionally” as “additionally,”; in line number 389, “the renal” as “renal”; in line number 393, “Institutional” as “the Institutional”; in line number 413, “EDTA coated” as “EDTA-coated”; in line number 419, “microplate” as “a microplate”; in line number 433, “intensity” as “the intensity”; in line number 435, “though” as “through”; in line number 436, “of of” as “of”; in line number 437, “shown on” as “shown in”; in line number 445, “heat mediated” as “heat-mediated”; in line number 457, “real time” as “real-time”; in line number 459, “Invitrogen,Thermo” as “Invitrogen, Thermo”; in line number 475, “Sigma-aldrich” as “Sigma-Aldrich”; in line number 478, “Full sized” as “Full-sized”; in line number 486, “manufacturer's” as “the manufacturer's”; in line number 486, “40-folds” as “40-fold”; in line number 493, “Data was” as “Data were”; in line number 503, “and to” as “and”.

Author Response

Q1. The suggestion is not properly carried out and should be rectified. Check the abbreviations throughout the manuscript and introduce the abbreviation when the full word appears the first time in the abstract and the remaining for the text and then use only the abbreviation. For example, in line number 34, the short (Fe) form of iron is not given and similarly for  “transferrin saturation” in line numbers 106 and 424. These types of corrections are need to be carriedout properly for all other abbreviations used in the manuscript.

Response: We corrected abbreviations (Fe, Tf, TfR1, HAMP, FPN, TSAT and WB) throughout the text. The corrections are shown in red color. We did not use abbreviations in the Abstract and Titles.

Comments on the Quality of English Language

Q1. There are some grammatical, alignment and typographical errors are noted in the manuscript and it should be thoroughly checked and corrected throughout the manuscript. For example, in line number 13, the word “kidneys” may be as “the kidneys”; in line number 13, “Iron” as “he iron”; in line number 15, “mouse” as “a mouse”; in line number 17, “transferrin” as “the transferrin”; in line number 23, “observasions” as “observations”; in line number 24, “potentailly” as “potentially”; in line number 26, “upregulation” as “the upregulation”; in line number 40, “that is” as “which is”; in line number 48, “incorporates in” as “incorporates”; in line number 49, “feroportin” as “ferroportin”; in line number 50, “all cells;, and is it consisted” as “all cells, and has or does it consisted”; in line number 80, “hephestin” as “hephaestin”; in line number 108, “transdermal” as “a transdermal”; all over the manuscript, “the Table” as “Table”; in line number 124, “reference” as “the reference”; in line number 163, “describe” as “described”; in line number 164, “Olympus” as “an Olympus”; in line number 187, “expression” as “the expression”; in line number 187, “renal” as “the renal”; in line number 188, “was analyzed” as “were analyzed”; all over the manuscript, “real time” as “real-time”; in line number 201 and 225, “PVDF” as “the PVDF”; in line number 207, “the prevent” as “prevent”; in line number 208, “in the circulation” as “to circulation”; in line number 222, “assessment” as “the assessment”; in line number 232, “but he” as “but the”; in line number 233, “oxidiaze” as “oxidize or oxidized”; in line number 258, “Hepcidin-Murine” as “the Hepcidin-Murine”; in line number 274, “negative” as “a negative”; in line number 284, “presence” as “the presence”; in line number 300, “renal” as “the renal”; in line number 311, “anti mouse” as “anti-mouse”; in line number 315, “ware” as “were”; in line number 345, “unknown” as “an unknown”; in line number 360, “explanation of” as “explanation for”; in line number 365, “normal” as “a normal”; in line number 385, “additionally” as “additionally,”; in line number 389, “the renal” as “renal”; in line number 393, “Institutional” as “the Institutional”; in line number 413, “EDTA coated” as “EDTA-coated”; in line number 419, “microplate” as “a microplate”; in line number 433, “intensity” as “the intensity”; in line number 435, “though” as “through”; in line number 436, “of of” as “of”; in line number 437, “shown on” as “shown in”; in line number 445, “heat mediated” as “heat-mediated”; in line number 457, “real time” as “real-time”; in line number 459, “Invitrogen,Thermo” as “Invitrogen, Thermo”; in line number 475, “Sigma-aldrich” as “Sigma-Aldrich”; in line number 478, “Full sized” as “Full-sized”; in line number 486, “manufacturer's” as “the manufacturer's”; in line number 486, “40-folds” as “40-fold”; in line number 493, “Data was” as “Data were”; in line number 503, “and to” as “and”.

Response: We corrected all noted grammatical, alignment and typographical errors. We also corrected “the Table” to “Table” and “real time” to “real-time” all over the manuscript.